# The Impact of AN Contribution on the Thermal Characteristics and Molecular Dynamics of Novel Acrylonitrile–Styrene–Styrene Sodium Sulfonate Terpolymers

**DOI:** 10.3390/polym13030420

**Published:** 2021-01-28

**Authors:** Hamud A. Altaleb, Abdullah M. Al-Enizi, Hany El-Hamshary, Sayed Z. Mohammady

**Affiliations:** 1Department of Chemistry, College of Science, King Saud University, Riyadh 11451, Saudi Arabia; haltaleb@iu.edu.sa (H.A.A.); amenizi@ksu.edu.sa (A.M.A.-E.); helhamshary@ksu.edu.sa (H.E.-H.); 2Department of Chemistry, Faculty of Science, Tanta University, Tanta 31527, Egypt; 3Chemistry Department, Faculty of Science, Cairo University, P.O. Box, Giza 12613, Egypt

**Keywords:** terpolymers, acrylonitrile, styrene, styrene sodium sulfonate, glass transition, thermal stability

## Abstract

We performed a free radical solution polymerization of new acrylonitrile (AN), styrene (St) and styrene sodium sulfonate (SSS) acceptor–donor acceptor monomer systems. The compositions and structures of the produced terpolymers were elucidated using CHNS elemental analysis, and Fourier transform infrared (FTIR) spectroscopies. Three terpolymers candidates were chosen for detailed thermal investigations, where the AN molar ratio varied almost threefold (from ~6.9% to ~17.4%) while the molar ratios of St and SSS varied slightly, at average values around 76.0% and 12.9%, respectively. The glass transition (T_g_) values of the terpolymers were measured calorimetrically. In addition, thermal gravimetric analyses (TGA) of the samples were conducted in the temperature range from room temperature to 800 °C. All terpolymers exhibited a single T_g_ value, indicating random copolymerization of the monomeric species. TGA results revealed that variation of the AN molar ratio had a significant influence on the thermal stabilities of the terpolymers. The impact of AN contribution on the molecular dynamics of the glass transition in the terpolymers was explained quantitatively in a framework of a molecular model.

## 1. Introduction

The preparation of multi-component polymers is a well-established method for producing polymers with chemical and physical properties for pre-defined purposes. For example, we can control the determination of elasticity by a calculated modification in the design of these polymers. This is often done by using additional monomers with different characteristics, which may be two or more [1,2], in polymerization. The co-polymer composition is the most influential factor on the final properties and depends on the participation ratios of the monomers involved in the resulting polymer. Moreover, the reactive ratios are also important for predicting the microstructure, polymerization rate, and hence the molecular weight distribution of polymers.

Despite the richness of the literature on the expansive and detailed information on the co-polymerization processes of two-monomer systems, when we compared the material, we found that a very limited number of studies have been devoted to studying the preparation and characterization of polymers. A detailed study of the polymerization process of butyl acrylate/methyl methacrylate/vinyl acetate (BA/MMA/VAc) [3,4,5,6] was performed. Co-polymerization was performed for the following pairs of monomers: BA/MMA, BA/VAc, and MMA/VAc. Studies have demonstrated that the produced polymers are distinguished by conversion, structure, and molecular weight. Estimates of co-polymerization reaction ratios were calculated. One of the most important results of this work is to reach an accurate prediction of the copolymer formation through the full affinity range of each of the said pairs of monomers. The study proved that the polymer has a very high molecular weight (this is evidenced by the high degree of branching in these copolymer systems).

Radical polymerization was performed on pair-type donor acceptor monomers, such as styrene (St), maleic anhydride (MA), and N-vinyl pyrrolidone (NVP) [7] in methyl ethyl ketone (MEK) under ray–ray at room temperature. The constants for co-polymerization, the complex formation, and some kinetic variants of the monomer systems were studied by UV spectroscopy and 1H NMR. The results obtained showed that the polymerization was mainly carried out through a “complex” mechanism between, for example, St/MA and MA/NVP systems. An important example that has been extensively studied previously is the free radical polymerization process of three non-polymerizable monomers [8,9,10]. In these studies, the free radical polymerization of three different systems [9] with heterocyclic monomers has been described. All combinations of the systems examined in these studies can be viewed as systems containing one donor and two receptor monomers. The study proved that the binary polymerization process results in an alternation of co-polymers for the receptor and donor monomer units. The kinetics of the alpha-methyl styrene (AMS)/butyl acrylate (BA)/methyl methacrylate (MMA) system has been studied in detail by Leamen et al. [11]. The authors show that terpolymer systems typically do not have monomers or cases where depropagation was a significant concern and therefore, the Alfrey–Goldfinger model was sufficient to describe the system. A sensitivity analysis and benchmarking of a model presented by the authors have been done and the terpolymerization system has been studied for selective feed ratios with promising results.

On the other hand, an assay for the radiation-induced dispersion polymerization of styrene in a mixture of water and ethyl alcohol was performed by Zhongqing Hu [12] using a new “graded” co-polymer consisting of maleic styrene anhydride (St—MA) and maleic anhydride—N-phenylpyrrolidone complexes (MA—NVP) as an installer.

In this work, we aim to synthesize novel terpolymers of AN, St, and SSS. In addition, we will explain the influence of the molar ratio of AN on the thermal properties of the resulting terpolymers. Moreover, the impact of the variation of AN as well as the donor acceptor ratio on the molecular dynamics of the glass transition in the terpolymers will be explained in a framework of the meander model of polymers. For this reason, three samples were chosen in this work such that the proportion of St as well as SSS monomers were retained without a significant change, while the molar ratio of AN varied approximately 2.5 times from about 6.9 to ~17.4%.

## 2. Materials and Methods

### 2.1. Materials

2,2′-Azobis(2-methylpropionitrile), AIBN, of 98% purity was purchased from the Sigma Aldrich Chemical Company. Acrylonitrile (AN) monomer and styrene (St) with 99% purity were purchased from TCI Company, Tokyo, Japan. Para-styrene sodium sulfonate (SSS) salt was purchased from Santa Cruz Biotechnology, and Dimethyl sulfoxide (DMSO) having a purity higher than 99% was purchased from Bio basic Inc, Toronto, Ont Canada.

All monomers were purified by distillation prior to the polymerization processes according to the boiling point of each monomer. Azobisisobutyronitrile (AIBN) was crystallized twice from ethanol and used fresh in the polymerization process.

Fourier transform infrared spectroscopy (FTIR) was performed on a spectrometer (Shimadzu, model: IRAffinity, Tokyo, Japan) in order to characterize the presence of the different specific functional groups in the terpolymers. Samples were prepared as KBr pellets. Measurements were carried out at room temperature in the range from 4000 to 400 cm^−1^.

Thermal stability of the polymers was investigated using thermogravimetric analysis (TGA) which was measured (Mettler Toledo, model: TGA/DSC1, Greifensee, Switzerland) under a nitrogen flow rate of 60 mL/min, in the temperature range from ambient temperature to 800 °C with heating rate of 10 °C/min.

Glass transition temperatures, T_g_, were determined using differential scanning calorimetry (DSC) utilizing a Shimadzu DSC-60A, Japan. Specimens of the size 2–3 mg were encapsulated in aluminum pans and were heated or cooled under a dry nitrogen atmosphere. Measurements were performed at four heating rates, namely 2.5, 5.0, 10.0, and 20.0 °C/min. Samples were heated from room temperature to 200 °C and cooled back to room temperature at the same heating rate of each particular experiment. Finally, samples are reheated again to 300 °C, all under an inert atmosphere of nitrogen gas. The T_g_ values were determined by the enthalpy inflection point in the second heating run. The accuracy of temperature monitoring was better than 1.0 °C.

### 2.2. Terpolymer Synthesis

Sequential polymerization steps were used for the synthesis of random free radical solution polymerization by varying the feed ratio. The recipe ratio and yield of all the polymers are listed in Table 1. The free radical polymerization of the three monomers AN, SSS, and St at a particular feed ratio was carried out in a 50 mL flask, condenser, and argon inlet. A volume of 25 mL of DMSO was added to a flask and purged with argon. The monomers and AIBN were then added to the flask. The polymerization reactions were performed at 60 °C in an oil bath for 24 h. After achieving the needed conversion (by trial and error), clearly viscous solutions of the terpolymers were obtained. To remove the residual monomers and any unreacted initiator species, the resultant viscous solutions were precipitated twice in diethyl ether and then left to dry in a vacuum at 50 °C for a period of 72 h. A possible scheme representing the chemical reaction involved in the synthesis of the terpolymers is shown below (Scheme 1).

## 3. Results and Discussion

### 3.1. FTIR and Elemental Analysis

From now on, we will refer to the three terpolymer samples in this work with symbols Ter-I, Ter-II, and Ter-III for the samples possessing AN molar ratios of 6.9, 9.1, and 17.4% (Table 2), respectively. A representative example of FTIR spectra of the terpolymers that showed characteristic absorbance in the range at 400–4000 cm^−1^ for the Ter-III sample is depicted in Figure 1. Absorption bands of CH_2_ asymmetric and symmetric stretching vibration were revealed at 2941 and 2835 cm^−1^_,_ respectively. The peaks at 2246 and 1430 cm^−1^ correspond to the (CN) of the AN monomers participating in the terpolymer chains [13]; however, the stretching vibrations of the sulfonate group attached to the phenyl ring of SSS is located at 1030 and 1117 cm^−1^, with increasing peak intensity with increasing SSS content [14]. The absorption maximum at 3413 cm^−1^ belongs to the OH stretching vibration, which can be attributed to the moisture, and the broad band becomes sharper with the increase in the sodium styrene sulfonate fraction.

### 3.2. Thermogravimetric Analysis (TGA)

Figure 2a,b depict the thermogravimetric (TGA) and the differential thermogravimetric analysis (DTGA) results of samples Ter-I, Ter-II, and Ter-III, respectively. Samples were subjected to a preheating isothermal step at 120 °C for approximately 20 min directly prior to the measurements in order to prevent the loss of humidity which usually happens at temperatures below ~170 °C. There was a common mass loss transition at ~170 °C for the three samples. The mass loss increased rapidly in the range from 220 to 230 °C (the point at which the thermal degradation rate was maximum). Then, the mass loss slowed. The mass loss that accompanied this step was 11.8% for Ter-I and 14.3% for both the Ter-II and Ter-III samples. This transition step could be rationalized to the decomposition of thermal weak bonds and some terminal end groups such as tail-to-tail or head-to-head bonds. Some contribution may originate from some initiator fragments within the polymeric chains [15,16,17,18].

Figure 2a,b reveals a second thermal degradation step in the temperature range from 350 to 490 °C. The mass loss that accompanied this step was 56.9%, 39.0%, and 39.9% for the Ter-I, Ter-II and Ter III samples, respectively. The thermal degradation rate of the second transition was maximum in the temperature range from 401.0 to 415.6 °C (Table 3). This step represents the main degradation step. It can be explained by random chain scission that can take place, resulting in a material having a shorter chain length. In a more advanced stage of the pyrolysis reaction, these shorter chains decompose into simple substances that are easy to volatilize at high temperatures. The shape as well as the position of this degradation step is quite similar to the polystyrene thermogram [19,20].

A third transition is recognized in Figure 2a,b. Its onset is slightly above 480 °C and the offset is ~600 °C. The mass loss observed in this step was ~3.0–6.0% for the three samples. This degradation step can be attributed to the pyrolysis of the sulfonate groups [20].

Finally, a fourth degradation step started to appear at ~600 °C until complete degradation was observed in all samples. This degradation step is attributed to the evaporation of polymer chain fragments and the concomitant complete decomposition due to liberation of gases such as NH_3_, HCN, and CO_2_ [15,16,17,18]. All degradation steps, the corresponding weight loss, and the peak maxima are summarized in Table 3.

Generally, it can be recognized that the order of the thermal stability of the samples was Ter-III > Ter-II > Ter-I. The remaining weight fractions of the samples at 600 °C were 28.3%, 39.0%, and 40.2% for Ter-I, Ter-II and Ter-III, respectively. This implies that the thermal stability increased with increasing AN content. Increasing the AN molar ratio will increase the total fraction of electron acceptor participating monomers within the chains (AN and SSS). The ratios of the electron donor (St) to electron acceptor monomers were 1:4.15, 1:3.57, and 1:2.50, for the Ter-I, Ter-II and Ter-III samples, respectively. That is to say, the thermal stability of the terpolymers increased when the donor acceptor monomer contributions were close to each other, which could be explained by the increase in the degree of stacking of polymeric chains that prohibited the degradation processes.

### 3.3. X-ray Diffraction (XRD)

X-ray diffraction measurements were performed by scanning the rotating sample in a wide range of angles (5–60°) for untreated polymers. The intensity was plotted against the diffraction angle of the samples as shown in Figure 3. The Bragg equation for diffraction [21] was applied to calculate the peak maxima values, which were found to be identical for the different samples as shown in Figure 3. The average interchain separation distance (d) of the terpolymers, evaluated from the strong maximum in the XRD scans, was found to be ~5.7 Å in the three samples. It is interesting to obtain such values of “d”. Higher “d” values were expected due to the SSS monomer contribution in terpolymer chains. The incorporated bulky sulfonate groups were expected to increase in the d values in the terpolymers. The obtained d values suggest that a two-dimensional plane π-π stacking between the terpolymer chains would take place and the sulfonate groups are dangled out of this plane.

### 3.4. Differential Scanning Calorimetric (DSC) Results

Figure 4 displays the differential scanning calorimetric thermograms of sample Ter-I at four different heating rates, namely, 2.5, 5.0, 10.0, and 20 °C/min. The temperature dependence of heat capacity exhibited a single common glass transition temperature, indicating random copolymerization of the three monomers within the terpolymer chains. According to Kuo et al. [22], in some cases, the monomer distribution might take place in tight blocks in the range of 10 to 30 nm [22]. It can be observed that the T_g_ values systematically shifted to higher temperatures at elevated heating rates. Table 4 represents the evaluation results of the T_g_ values of the particular terpolymers at different heating rates.

It is obvious that as the AN content increased the T_g_ of the terpolymers shifted systematically to higher temperature values. This can be explained by the increase in the strength of the hydrogen bonding in the system when the AN molar ratio increased, where AN can be considered a strong acceptor with styrene donating monomers. The T_g_ composition dependence will be explained more quantitatively later in the frame of the meander model [23,24,25,26,27,28,29].

### 3.5. Activation Curves and the Cooperative Rearranging Regions of the Glass Process

It is well established that the activation curves of the glass process do not exhibit linear dependency on the reciprocal of absolute temperature. In other words, the kinetics of the glass process do not obey the Arrhenius law presented in Equation (1), which is
(1)$$f_{\max}=\frac{f_{0}}{\pi }e^{\frac{-E_{a}}{kT}}$$

*f*_max_ refers to the relaxation frequency of the maximum, *k* is the universal gas constant, *f*_0_ assigns the segmental vibrational frequency at infinite temperature, *T* is the absolute temperature, and *E_a_* is the activation energy. It should be noted here, that if the Arrhenius equation is tested on experimental data at temperatures slightly above T_g_, then the obtained parameters (local vibration frequency and activation energy) and their values cannot be explained by molecular data. This conclusion may be then taken as n evidence of the cooperativity of the glass process.

A remarkable effort has been exerted to explain the collaborative nature of the glass process in a quantitative manner. As a result, it is now well established that there is a region within which cooperative particle motion can arise and that decreases in size with increasing temperature. This domain has been termed the cooperative rearrangement region (CRR) [23,24,25,26,27,28,29,30,31]. Following the information on this topic, the available CRR values can range between 2 and 25 nm, depending on the method of treating and interpreting data via different models [23,24,25,26,27,28,29,30,31].

An empirical formula was obtained by Stoll and coworkers that enabled them to correlate the glass transition temperature estimated by conventional DSC curves to the relaxation time corresponding to the enthalpy relaxation:(2)$$q\cdot \tau_{veq\;}=15K$$
where q is the calorimetric cooling or heating rate and $\tau_{veq}$ is the relaxation time of enthalpy at T_g_. Merging Equation (2) with the well-known relationship that correlates the relaxation time and the maximum frequency of relaxation in dynamic experiments, the following can be obtained: [32,33]:
(3)$$2\pi f_{v}\tau_{veq}=1$$

A relaxation frequency can be predicted at T_g_ which has been evaluated from conventional DSC measurements. This relaxation frequency is equivalent to the frequency of the maximum of the imaginary part in modulated DSC experiments.

Figure 5 depicts the activation curves of the terpolymer samples delivered from the DSC measurements at different heating rates after the use of Equations (2) and (3).

Activation curves in Figure 6 can be fitted by means of the well-known VFT equation or its mathematical equivalent WLF, equation [34].
(4)$$\log f_{\max}=\log f_{0}-\frac{C_{1}(T-T_{0})}{C_{2}+(T-T_{0})}$$

*C*_1_ and *C*_2_ in Equation (2) are fitting parameters, while *T_0_* is a constant value normally taken in the neighborhood of T_g_. Actually, these equations cannot provide any useful information about CRR. However, Pechhold [23,24,25,26,27,28,29] proposed a ‘dislocation concept’ for bundles of polymer molecular chains. The smallest isotropic arrangements of these bundles may be considered as cooperatively rearranging units (CRRs) [23,24]. Furthermore, the diameter (r) of the explained molecular bundles was calculated by thermodynamical free energy minimization treatments [23,24].

Moreover, this r was modified by the superfolding mechanism proposed in the meander model, ultimately producing a nine-fold cube of edge length (*L*) with *L* = 3r. On the other hand, Pechhold [23,24,25,26,27,28,29] suggested the following equation (Equation (5)) for treating the activation behavior of the glass process:
(5)$$f_{\max}=\underline{\underline{\frac{f_{0}}{\pi }e^{\frac{-Q_{\gamma }}{kT}}}}\cdot \underline{\underline{{\left[1-{\left(1-e^{\frac{-\varepsilon_{s}}{kT}}\right)}^{\frac{L}{d}}\right]}^{3{\left(\frac{L}{d}\right)}^{2}\frac{d}{s}}}}$$
$$\begin{array}{cc} ⇑ & ⇑\\ \mathrm{Arrhenius factor} & \mathrm{cooperatively factor} \end{array}$$
where *L* is the length of the CRR cube, *f*_0_ is the local vibrational frequency, *d* is the interchain distance, *s* is the average length of the segment, *ε_s_* is the energy of dislocation, and *Q_γ_* is the activation energy of the local segmental motion. The *d* values for terpolymer samples were determined by powder X-ray diffraction (Figure 3). The average value of *s* was 0.47 nm for terpolymer samples. Fitting results of Equation (5) are presented in Figure 6a–c.

The parameter *ε_s_* increases significantly with the increase in the AN content (Figure 6a). In other words, *ε_s_* increases as the molar ratio of the acceptor to donor monomers approaches 1. The acceptor to donor ratio may cause a better packing of the molecular chains of the terpolymer through the expected *π-π* stacking. The four *π* electrons in the nitrile group of the AN monomer are expected to enhance the *π-π* stacking. Consequently, the probability of dislocation is hindered. Therefore, both values of *ε_s_* and T_g_ increase. The evaluated *L* values show good agreement with the experimental results obtained from the DSC, where the T_g_ values move to higher temperatures with the increase in the CRR length (*L*). The pronounced variation of *L* values with the AN content (Figure 6b) can be attributed to the increase in the terpolymer chain stiffness as the monomer fraction of AN increases. The strengthening in stiffness of the terpolymer chains will limit the ability of folding as well as superfolding mechanisms (in order to build the bundles and meander cubes), leading to high values of bundle diameters. The final result of this is the increase in the size of the meander cubes (*L*^3^). The increase in *Q_γ_* values (Figure 6c), reflecting the temperature dependence of *f*_max_ at elevated temperatures, when the AN fraction increases explains the high stiffness of the terpolymers chains even at increased temperatures away from T_g_.

## 4. Conclusions

Three new terpolymers of acrylonitrile (AN), styrene (St), and styrene sodium sulfonate (SSS) acceptor-donor acceptor monomer systems were prepared and investigated. The AN molar ratio varied almost two and a half times (from 6.9% to 17.4%) while the molar ratios of St and SSS were retained at average values around 76.0 and 12.9%, respectively. TGA results revealed all terpolymers exhibited four thermal degradation peaks. The variation of the AN molar ratio and the donor acceptor ratios of monomers had a significant impact on the position of the degradation peaks assuming a direct influence of the AN mole fraction on the entire temperature range representing the thermal stabilities of the terpolymers.

The glass transition (T_g_) values of the terpolymers were measured calorimetrically at four different heating rates. All terpolymers exhibited a single common T_g_ value indicating random copolymerization of the monomeric species. The results were further discussed in the framework of the meander polymer model. The evaluations showed that the parameter *ε_s_* increased significantly with the increase in the AN content and as the molar ratio of the acceptor to donor monomers approached 1. The four *π* electrons in the nitrile group of the AN monomer are suggested to enhance the π-π stacking between the terpolymer chains. The evaluated *L* values showed good agreement with the experimental results obtained from the DSC, where the T_g_ values moved to higher temperatures with the increase in the CRR length (*L*). The marked variation of *L* values with the AN content was attributed to the increase in the terpolymer chain stiffnesses as the monomer fraction of AN increased. The strengthening in the stiffness of the terpolymer chains limited the ability of folding and superfolding mechanisms, leading to high values of bundle diameters. The final result of this was the increase in the size of the meander cubes (*L*^3^). The elevated *Q_γ_* values with the increase in the AN fraction explains the high stiffness of the terpolymers chains even at increased temperatures away from T_g_.

## Data Availability

Not applicable.

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
