# Peer review of "The Impact of AN Contribution on the Thermal Characteristics and Molecular Dynamics of Novel Acrylonitrile–Styrene–Styrene Sodium Sulfonate Terpolymers"

_polymers, 2021, doi:10.3390/polym13030420_

Round 1

Reviewer 1 Report

This article reports the experimental studies on the synthesis of novel terpolymers. The influence of the molar ratio of acrylonitrile on the thermal properties of the resulting terpolymers is elucidated. The chosen subject is of interest from both fundamental and practical points of view. In my view, the paper provides relevant information and is well written. I believe it could be published after some text editing.

Author Response

First Reviewer:

- Comment  no. 1: It could be published after some text editing.

Text editing has been done in the revised manuscript

Reviewer 2 Report

The article entitled "The impact of AN contribution on the thermal characteristics and molecular dynamics of novel acrylonitrile-styrene-styrene sodium sulfonate terpolymers" describes the synthesis and characterization of a new type of terpolymers. While the research may be of interest to the reader, this article requires substantial modification before it is accepted for publication in Polymers. Below are the attached comments, final to be considered in the revision process.

- Chapter numbering and text formatting must be significantly improved during the revision process. I am also asking for the correct numbering of the drawings. The article should be significantly improved in terms of editing, as well as I recommend using language correction, taking into account an English native speaker.

- Figure showing the chemical structure of terpolymers should be corrected by adding n value in the bottom index of the square bracket.

- The DTG curves should be presented as part of the thermal stability analysis. Besides, the ending of the curves raises concerns. They look as if they tended to values ​​below 0%, which would indicate an incorrect measurement; please comment or correct the TG curves. The characteristic values ​​for the thermal decomposition curves, including the DTG peak values, should be presented in a separate table.

- Please provide FTIR spectra for all types of terpolymers along with the improvement of the graph quality. The tests were performed in transmission, so it is possible to perform a quantitative analysis of the differences between the materials. In addition, detailed information on the FTIR measurement parameters should be provided in the methodology section.

- Referring to the TGA thermograms, it can be noticed that below the temperature of 200°C a decrease in mass was noted. The weight loss due to the presence of water is usually observed up to the value of 130°C. Therefore, the observed changes may indicate the polymer's degradation or the residues of low-molecular fractions after synthesis. This has to be ruled out; otherwise, the DSC tests may be affected by the degradation occurring during the first heating to 200°C. Please provide in response and/or supplementary data curves representing the first heating. The methodology for determining the Tg was not given; therefore, relevant information should also be provided.

- NMR analysis should be significantly expanded and discussed more deeply in order to confirm the structure of terpolymers.

- The authors wrote: "It can be observed that the Tg values ​​are systematically shifted to higher temperatures at elevated heating rates." It should be remembered and mentioned that this phenomenon is mostly an apparatus effect.

Author Response

Second Reviewer:

Comment  no. 1:

- Chapter numbering and text formatting must be significantly improved during the revision process.

Chapter numbering and text formatting have been corrected.

- I am also asking for the correct numbering of the drawings.

Figures numbers are corrected.

- The article should be significantly improved in terms of editing, as well as I recommend using language correction, taking into account an English native speaker.

The article has been sent for editing and proofreading (please see the attached certificate).

Comment  no. 2:

- Figure showing the chemical structure of terpolymers should be corrected by adding n value in the bottom index of the square bracket.

Correction has been done and the “n” value is added in the right place.

Comment  no. 3:

- The DTG curves should be presented as part of the thermal stability analysis. Besides, the ending of the curves raises concerns. They look as if they tended to values below 0%, which would indicate an incorrect measurement; please comment or correct the TG curves. The characteristic values for the thermal decomposition curves, including the DTG peak values, should be presented in a separate table.

The DTG results are now presented in Figure 2b. No negative values present in Figure 2a. The remaining weight reaches small values above 750 oC; but still above zero. All the characteristic values of the degradation processes are now included in Table 3.

Comment  no. 4:

- Please provide FTIR spectra for all types of terpolymers along with the improvement of the graph quality. The tests were performed in transmission, so it is possible to perform a quantitative analysis of the differences between the materials. In addition, detailed information on the FTIR measurement parameters should be provided in the methodology section.

FTIR spectra of all terpolymer samples are now depicted in Figure 1. No quantitative analysis can be performed unless exact equal amounts and sample thicknesses are used. For this reason, FTIR is usually used as a qualitative tool of analysis. Actually, elemental analysis is more powerful in determination of the contribution of each monomer in the terpolymers. More details about the FTIR experiments have been involved in the methodology section.

Comment  no. 5:

- Referring to the TGA thermograms, it can be noticed that below the temperature of 200°C a decrease in mass was noted. The weight loss due to the presence of water is usually observed up to the value of 130°C. Therefore, the observed changes may indicate the polymer's degradation or the residues of low-molecular fractions after synthesis. This has to be ruled out; otherwise, the DSC tests may be affected by the degradation occurring during the first heating to 200°C. Please provide in response and/or supplementary data curves representing the first heating.

It can be noticed in Figure 2a that the mass loss at 200°C is less than 3% in all terpolymers samples. The heating rate involved in Figure 2a is 10°C /min. It should be taken in consideration that the thermal degradation curves as well the differential curves always shift to higher temperatures upon increasing the heating rate. That is to say, the degradation temperature at 20 oC/min must be higher and the results in Figure 4 will not be affected [1].

- The methodology for determining the Tg was not given; therefore, relevant information should also be provided.

It is mentioned in the methodology section: “The Tg values were determined by the enthalpy inflection point in the second heating run”.

Comment  no. 6:

- NMR analysis should be significantly expanded and discussed more deeply in order to confirm the structure of terpolymers.

There is overlapping between the HNMR spectra of the aromatic protons in styrene (St) and styrene sodium sulfonate (SSS). In addition, the HNMR bands corresponding to the chemical shift of the proton bonded to the carbon atom attached to the cyano groups in acrylonitrile segments overlaps also with the protons of water which is difficult to be excluded from SSS segments (highly hygroscopic). Due to these difficulties, we decided to exclude the HNMR data. Instead, we were content with the elemental analyses results in determining the terpolymers composition.

Comment  no. 7:

- The authors wrote: "It can be observed that the Tg values are systematically shifted to higher temperatures at elevated heating rates." It should be remembered and mentioned that this phenomenon is mostly an apparatus effect.

It is well established that Tg is time dependent process, i.e., it is dependent on the rate at which it is measured. Whatever the method performed to determine Tg, its values are shifted to higher temperatures at elevated rates (heating rates in DSC). 

[1] Reference no. 16 in the manuscript “Al-Rasheed, H.H.; Mohammady, S.Z.; Dahlous, K.; Siddiqui, M.R.H.; El-Faham, A. Synthesis, characterization, thermal stability and kinetics of thermal degradation of novel polymers based-s-triazine Schiff base.  J. Polym. Res. 2020, 27, 10.”